# Storage Stability and Disinfection Performance on *Escherichia coli* of Electrolyzed Seawater

**Regina G. Damalerio [1], Aileen H. Orbecido [1], Marigold O. Uba [2], Patricio Elvin L. Cantiller [2] and Arnel B. Beltran [1],***

[1] Chemical Engineering Department, De La Salle University, Malate, Manila 1004, Philippines; regina_damalerio@dlsu.edu.ph (R.G.D.); aileen.orbecido@dlsu.edu.ph (A.H.O.)

[2] Biology Department, De La Salle University, Malate, Manila 1004, Philippines; marigold.uba@dlsu.edu.ph (M.O.U.); patricio.cantiller@dlsu.edu.ph (P.E.L.C.)

* Correspondence: arnel.beltran@dlsu.edu.ph; Tel.: +632-524-4611 (local 218)

**Abstract:** The study investigated the effect of storage conditions on the stability of electrolyzed seawater (ESW)'s physicochemical properties (pH, oxidation-reduction potential (ORP), and free chlorine (FC) concentration), and bactericidal efficiency on the fecal coliform *Escherichia coli* for 30 days. Preliminary experiments were conducted to determine the optimal current and electrolysis time. Two batches of 2750 mL filtered seawater were electrolyzed using 50 mm × 192 mm platinum–titanium mesh electrodes at a current of 1.5 A for 20 min. One hundred milliliters of electrolyzed solution was transferred into each amber glass and high-density polyethylene (HDPE) bottles. The bottles were stored in a dark area at ambient temperature. The results showed an increase in pH and a decrease in ORP and FC concentration through time. Hypochlorous acid remained as the dominant component since the pH levels of the solutions remained below 7.5. FC decay was investigated using Chick's Law. It was determined that the decay in HDPE bottles (k = −0.066 day$^{-1}$) was faster compared to amber glass bottles (k = −0.046 day$^{-1}$). Nonetheless, HDPE bottles could still be used as an alternative container for 30 days only due to observed instability beyond 30 days. ESW remained effective since no surviving population of *E. coli* was observed throughout the experimentation.

**Keywords:** disinfection performance; *Escherichia coli*; electrolysis; kinetics; storage stability; seawater

## 1. Introduction

Electrochemical disinfection was introduced as an alternative due to its advantages over utilization of chlorine reagents generated from chlorine gas [1–5]. It is performed by passing current through chloride-containing water, which can be made synthetically (brine solution) or collected from a natural source (seawater) [2,6]. The presence of sodium chloride (NaCl) in the solution is important for the formation of oxidizing agents, such as free chlorine (FC), sodium hypochlorite (NaClO), or hypochlorous acid (HOCl) [5,7]. These compounds are responsible for inactivating pathogenic micro-organisms by oxidizing or attacking both the inner and outer membrane of the cell [8–10]. Other advantages of electrochemical disinfection include on-site production, raw material availability, an environment-friendly process, a less-hazardous reaction, low cytotoxicity, and no reported microbial resistance [10]. Furthermore, it has been utilized in several sectors, such as agricultural and food industries, since the disinfectant can be produced over wide pH ranges [11–15]. Xie et al. [12] investigated the effect of acidic electrolyzed water stored at different temperatures on contaminated raw shrimp, while Kasai et al. [16] investigated the effect of electrolyzed seawater (ESW) in naturally contaminated oysters. Moreover, the disinfectant can be applied in sanitizing materials and equipment.

Osafune et al. [10] concluded that electrolyzed acidic water is effective as it destroyed the cellular structures of the three bacterial species cultivated from the kendo equipment.

Seawater has 3.5% salinity, which predominantly consists of sodium and chloride ions. Furthermore, it also contains bromine, which is converted into an oxidant through electrolysis. These oxidizing compounds are detrimental towards pathogenic micro-organisms [4,14,17,18]. Thus, ESW has been employed and studied as a disinfectant [9,14,16–19]. However, the main problem lies in its storage stability through time [1,2]. The disinfectant is considered stable when it is still effective for inactivating pathogenic microorganisms. Residual oxidants or FC remains present and dominant, despite the changes in the physicochemical properties and storage conditions. No studies related to storage stability of ESW can be found in the literature since it is more often viewed as an alternative to water disinfection and as a disinfectant to be utilized immediately. Therefore, there is a possibility that the production and utilization of the disinfectant will be maximized by investigating its storage stability through time. The study could benefit people living near coastal areas, where raw material is abundant and accessible. Furthermore, this could serve as temporary alternative solution when sanitation is inaccessible (e.g., calamities).

The storage stability of ESW is determined by investigating the effects of the storage conditions and changes in the physicochemical properties, such as pH, oxidation-reduction potential (ORP), and FC concentration through time, to its bactericidal efficiency on the fecal coliform *Escherichia coli*. The rate of FC decay in amber glass and high-density polyethylene (HDPE) bottles is investigated using Chick's Law to determine the possibility of using HDPE bottles as an alternative storage container to amber glass bottles.

## 2. Materials and Methods

### 2.1. Materials

Glassware, platinum–titanium mesh electrodes, and reagents, such as *N*,*N*-dimethyl-p-phenylenediamine (DPD) powder, phosphate-buffered saline, nutrient agar, and broth powder, were purchased from several local suppliers. Purified stock culture of *E. coli* was acquired from the microbiology laboratory. The seawater sample utilized in the experiment was collected five meters away from the beach shore of Oriental Mindoro, Philippines. It was filtered using a PM10 microfilter before being stored at ambient temperature.

### 2.2. Electrolysis Setup

Fifty by 192 mm titanium–platinum mesh electrodes (mesh size: $10 \times 5$ mm; electrode surface area = 23 cm$^2$) were sonicated for 30 min to remove unwanted particles on the surface [18,19], and submerged vertically with a 0.5 cm gap in a 4 L Pyrex beaker. An alternating current (AC) power supply was utilized as an electrical source (Figure 1). Twenty each of 120 mL amber glass bottles and 250 mL HDPE bottles were cleaned and labeled prior to the day of the electrolysis experiment.

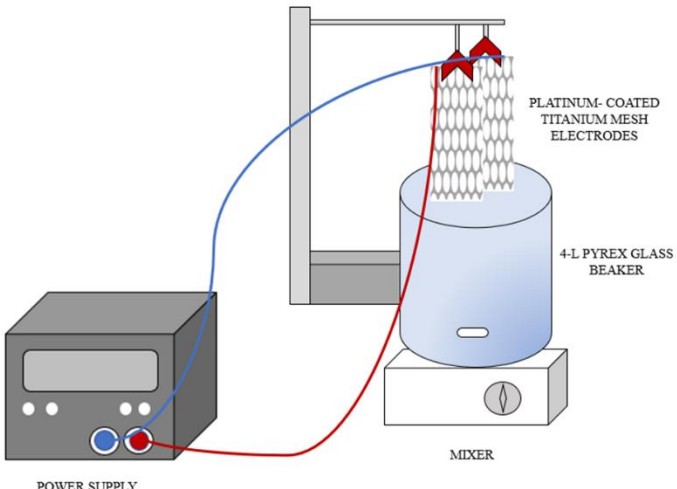

**Figure 1.** Electrolysis Setup.

## 2.3. Electrolysis of Seawater

Two batches of 2750 mL seawater were electrolyzed at a current of 1.5 A for 20 min. The applied current and electrolysis time were determined based on preliminary electrolysis experiments. It was carried out by investigating the effects of varying current and electrolysis duration on the physicochemical properties of ESW.

## 2.4. Storage Experiment

About 100 mL of ESW was transferred into each amber glass and HDPE bottle. Bottles were stored in a dark place at ambient temperature. The physicochemical properties and bactericidal efficiency of ESW were evaluated for 30 days.

## 2.5. Preparation of the Working Culture of E. coli

A purified stock culture of *E. coli* was cultivated every two weeks with nutrient agar and maintained at 4–10 °C for the duration of the study. A working bacterial suspension, prepared from the stock culture, was standardized prior to every disinfection experiment. This was performed by inoculating fresh colonies of *E. coli* in sterile distilled water until the turbidity was adjusted to equal McFarland Standard #0.5, which approximates $1.5 \times 10^8$ colony forming units (CFU) per mL [20]. A standard plate count was also performed to confirm the microbial density of the standardized turbidity [1,2].

## 2.6. Disinfection Experiments

To determine the effect of ESW on *E. coli*, 1 mL of the standardized *E. coli* suspension was mixed with and exposed to 9 mL ESW for one minute. Then, 1 mL of the latter mixture was added to 9 mL of Phosphate-Buffered Saline for one minute to render the ESW ineffective. A 10-fold serial dilution was performed by adding 1 mL from the previous tube to 9 mL of sterile distilled water up to $10^{-3}$ dilution. Using the spread plate technique, 0.1 mL aliquot from each solution was inoculated on nutrient agar plates. These were performed in duplicates. After incubation at 37 °C for 18–24 hours [1,2], the plates were examined for growth and the CFU/mL of *E. coli* was determined.

## 2.7. Electrolysis Setup

The pH and ORP values were measured using a pocket pH meter (Lutron, PH-211) and an ORP meter (Lutron, ORP-213; Ag/AgCl reference electrode), and ranged from pH 0.00 to 14.00 and ORP 0.0 ± 1000.0 mV, respectively. The pH meter was calibrated using standard buffers at a pH of 4.0, 7.0, and 10.0. The *N,N*-dimethyl-p-phenylenediamine (DPD) colorimetric method employing a UV/Vis Spectrophotometer at a wavelength of 515 nm was used to determine the FC concentration [1].

## 3. Theoretical Framework

Due to limited literature data, the system is assumed to contain sodium chloride only. Equations (1–6) below show reactions occurring at the anode and cathode sides during electrolysis. Water and chloride ions at the anode are oxidized to oxygen, hydrogen ions, and gaseous chlorine, while water at the cathode is reduced to hydroxides and hydrogen gas. The evolution of chlorine becomes more favorable than oxygen at the anode side due to the increase in hydrogen ion concentration in the solution [6].

Anode Side

$$2H_2O \rightarrow O_2 + 4H^+ + 4e^- \qquad E_o = 1.23\ V \tag{1}$$

$$2Cl^- \leftrightarrow 2e^- + Cl_2(g) \qquad E_o = 1.36\ V \tag{2}$$

Cathode Side

$$2H_2O + 2e^- \rightarrow 2OH^- + H_2 \qquad E_o = -0.83\ V \tag{3}$$

Aqueous chlorine formed from gaseous chlorine partially dissociates to hypochlorous acid and chloride and hydrogen ions at increasing concentrations. A further increase in the concentration of hypochlorous acid would result in its decomposition to hypochlorite and hydrogen ions.

$$Cl_{2(aq)} + H_2O \leftrightarrow HOCl + Cl^- + H^+ \qquad K_H = 4.0 \times 10^{-4} \text{ at } 25\ °C \tag{4}$$

$$HOCl \leftrightarrow OCl^- + H^+ \qquad K_H = 3.0 \times 10^{-8} \text{ at } 25\ °C \tag{5}$$

The distribution of chlorine species can be determined from Equations (4) and (5) by expressing it as equilibrium expressions. Simplifying these expressions would result in two equations as functions of pH and/or chloride concentration. Equation 6 determines the ratio of aqueous chlorine to hypochlorous acid, while Equation (7) determines the ratio of hypochlorous acid to hypochlorite [1,21].

$$\frac{\left[Cl_{2\ (aq)}\right]}{[HOCl]} = 10^{pK_H - pH + \log Cl^-} \tag{6}$$

$$\frac{[OCl^-]}{[HOCl]} = 10^{pH + pKa} \tag{7}$$

The distribution of aqueous chlorine, hypochlorous acid, and hypochlorite ions is plotted in Figure 2. It is observed that aqueous chlorine is present in highly acidic conditions and decreases beyond pH 2.0. The concentration in an ideal system decreases as it approaches pH 4.0 and hypochlorous acid dominates the region beyond pH 4.0. Hypochlorous acid is the most important component for ESW due to its high oxidizing property, which is responsible for inactivating pathogenic micro-organisms [8]. Furthermore, it is more stable than aqueous chlorine due to its less volatile nature [1,22]. Hypochlorous acid decreases gradually due to its weak acidic nature and remains predominant until pH 7.5, as illustrated in Figure 2b. Beyond pH 7.5, hypochlorite ions become the predominant component of the solution.

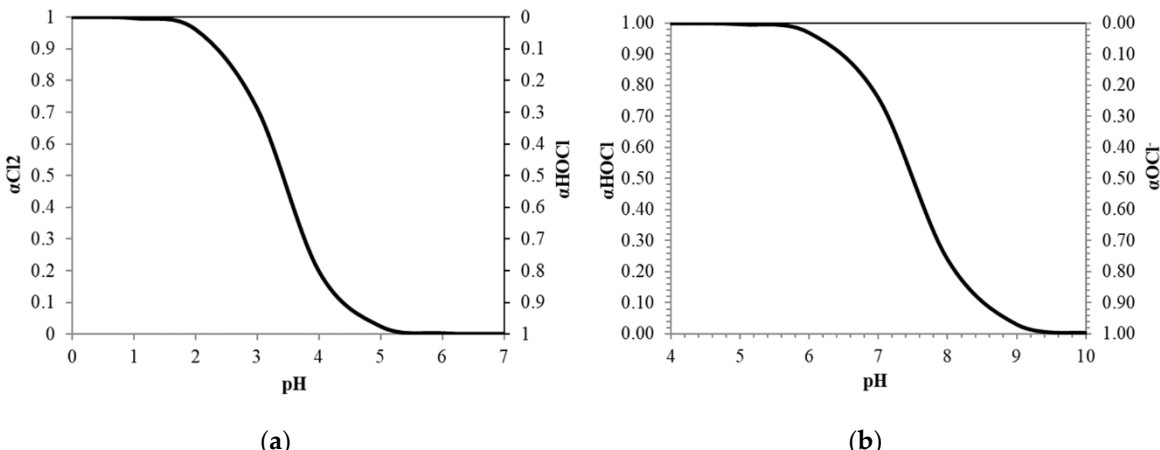

**Figure 2.** Distribution of (**a**) aqueous chlorine; and (**b**) hypochlorous acid in electrolyzed seawater (ESW) at different pH levels.

## 4. Results and Discussion

### 4.1. Electrolysis Parameter Selection

The preliminary experiments aided in determining the behavior of the physicochemical properties during the electrolysis of the seawater sample. It defined the relationship of the three properties and the effect of increasing applied current to the system through time. Figure 3a shows the decrease of pH at increasing electrolysis time and applied current. ESW becomes more acidic as the current increases due to continuous generation of hydrogen ($H^+$) ions. At the end of the electrolysis, the pH of ESW at 0.5 A, 1.0 A, and 1.5 A reached values of 5.76, 3.77, and 3.35, respectively.

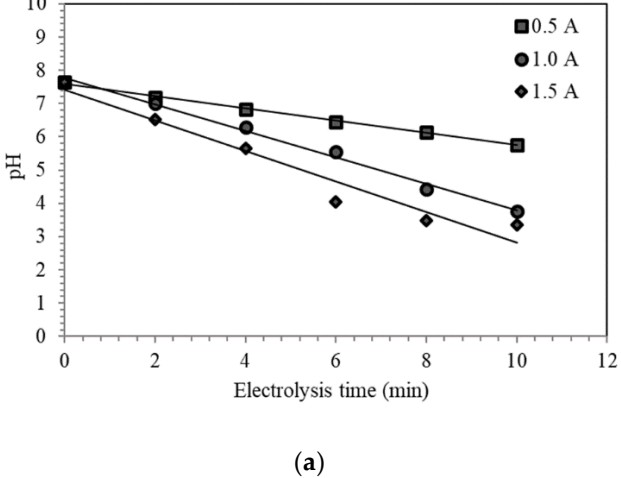

(**a**)

**Figure 3.** *Cont.*

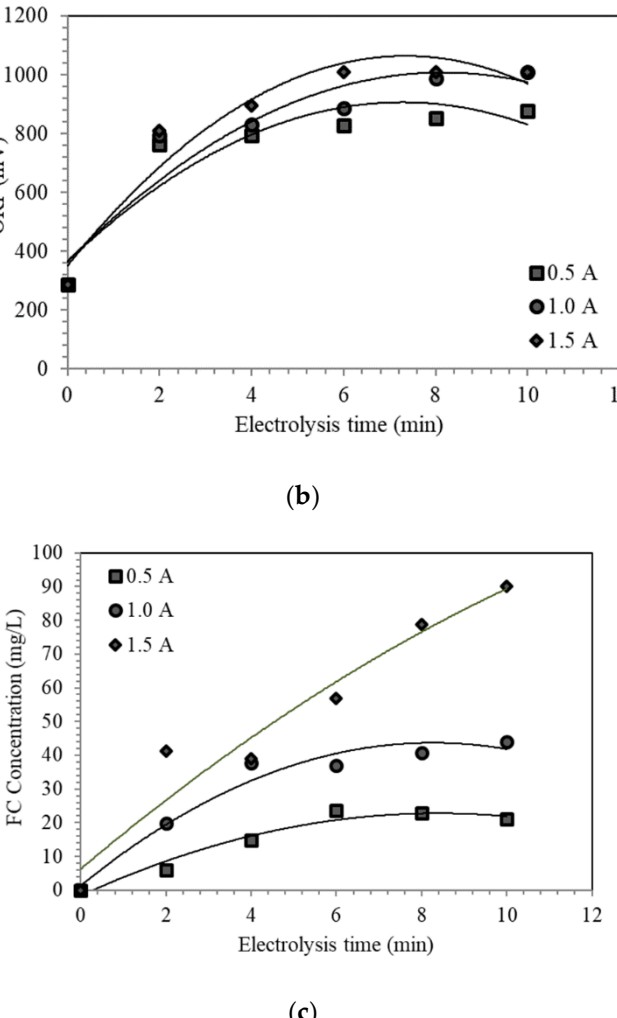

**(b)**

**(c)**

**Figure 3.** Preliminary results of ten minutes of ESW conducted at 0.5 A, 1.0 A, and 1.5 A for the following physicochemical properties: (**a**) pH; (**b**) oxidation-reduction potential (ORP); and (**c**) free chlorine (FC) concentration.

An increase in the acidity of the solution correlates to an increasing amount of oxidizing compounds, such as aqueous chlorine and hypochlorous acid, in the solution. Thus, high values of ORP and FC concentration are expected [1,2,12,13,18]. However, stability in ORP was observed from six minutes onwards at all currents, while stability in FC concentration was achieved from six minutes onwards at currents of 0.5 A and 1.0 A (Figure 3b–c). The stability of the FC concentration could mean that the maximum concentration of hypochlorous acid was attained at a given current. Based on Figure 2, electrolyzed solutions generated at currents of 0.5 A and 1.0 A fall in the region where hypochlorous acid is the predominant component. There would be minimal changes in the chlorine species due to a low concentration of aqueous chlorine and slow dissociation of hypochlorous acid to hypochlorite ions. The electrolyzed solutions at 1.5 A predominantly contain aqueous chlorine. Due to its volatile nature, some of the aqueous chlorine could have dissociated to hypochlorous acid to achieve equilibrium, as stated in Equation (6).

From the preliminary results, the current 1.5 A was utilized to determine the maximum electrolysis time to be used for the actual experimentation. Two runs of electrolysis were conducted and samples were collected every five minutes for evaluation of physicochemical properties. It is shown in Figure 4 that stability was achieved at an electrolysis time of 20 min (pH = 4.54, ORP = 988 mV, FC = 48.97 mg/L $Cl_2$). ESW generated at 20 min was determined to contain a maximum of 86.47% hypochlorous acid.

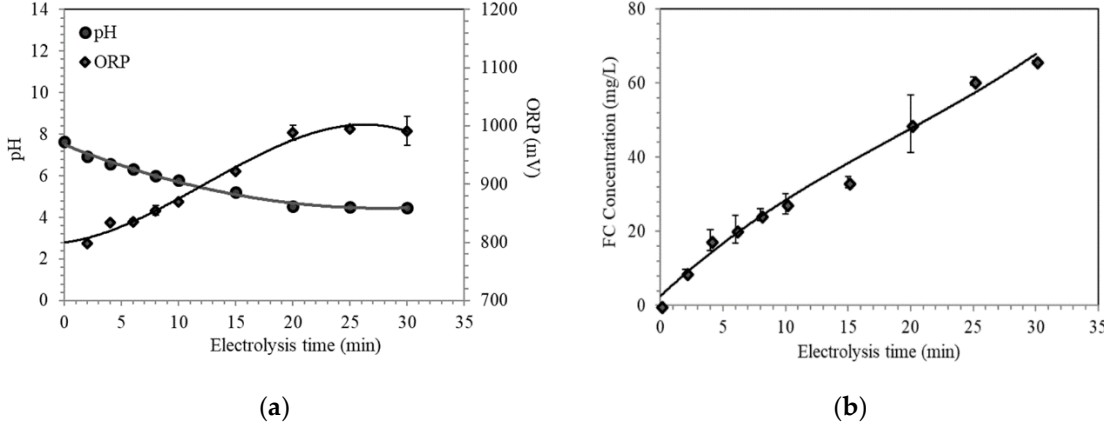

**Figure 4.** Behavior of (**a**) pH and ORP; and (**b**) FC concentrations at 1.5 A for 30 min of electrolysis.

The power required for ESW generated at a current of 1.5 A for 20 min was determined using the formula [4]:

$$P = V \times I \tag{8}$$

where:

P: Power requirement, watts
V: Observed voltage, volts
I: Current applied, ampere

The energy consumption can be calculated by multiplying the calculated power and electrolysis time [4]:

$$E = P \times t = V \times I \times t \tag{9}$$

where:

E: Energy consumption, watt-hour
P: Power requirement, watts
V: Observed voltage, volts
I: Current applied, ampere
t: Electrolysis time, hours

The calculated power required for generating one liter of ESW is 2.62 watts. If the system is upscaled to produce 1 kiloliter of the disinfectant, the power requirement is about 2620 watts (2.62 kW). On the other hand, the energy consumption of 1 kiloliter of electrolyzed seawater is about 873 watt-hours.

### 4.2. Storage Effect on Physicochemical Properties

#### 4.2.1. Evaluation for 30 Days

Figure 5 illustrates the behavior of pH and ORP of stored ESW in amber glass and HDPE bottles throughout the 30-day storage. Initially, the ESW in amber bottles (pH = 4.21, ORP = 1010 mV) contained 83.71% hypochlorous acid, while the ESW in HDPE bottles (pH = 5.74, ORP = 920 mV) contained 96.86% hypochlorous acid. It can be observed that there is a gradual increase in pH and a decrease in ORP throughout the storage period. Slow changes in pH and ORP could be attributed to the nature of the acid and storage conditions that minimized decomposition of hypochlorous acid through time.

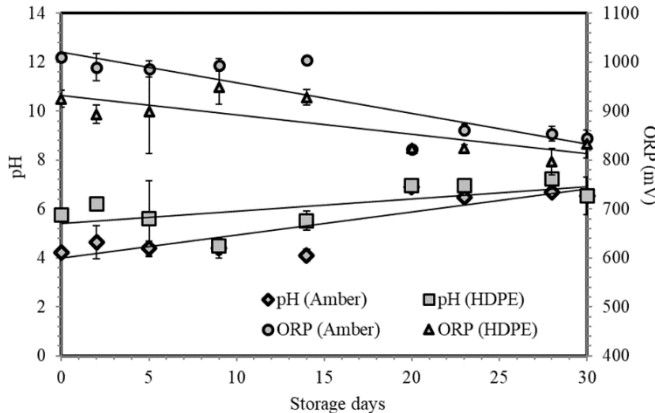

**Figure 5.** The pH and ORP distribution of ESW stored in both bottles throughout the 30-day storage period.

An average pH of 6.53 was measured for 30-day old solutions stored in both bottles. There was a 2.39% increase of hypochlorous acid in amber glass bottles and an 11.51% decrease in HDPE bottles. The increase of the hypochlorous acid in amber glass bottles may be attributed to partial dissociation of remaining aqueous chlorine. On the other hand, the decrease of ORP from 845.50 to 833 mV, for both bottles, could be due to the increasing decomposition of hypochlorous acid [1].

The average initial FC concentrations of ESW are 51.85 mg/L for amber glass bottles and 51.23 mg/L for HDPE bottles (Figure 6). Unlike in pH and ORP, FC in both bottles experienced an exponential decline through time. By day 30, approximately 22.72% and 8.80% of the initial FC concentration remained in amber glass and HDPE bottles, respectively.

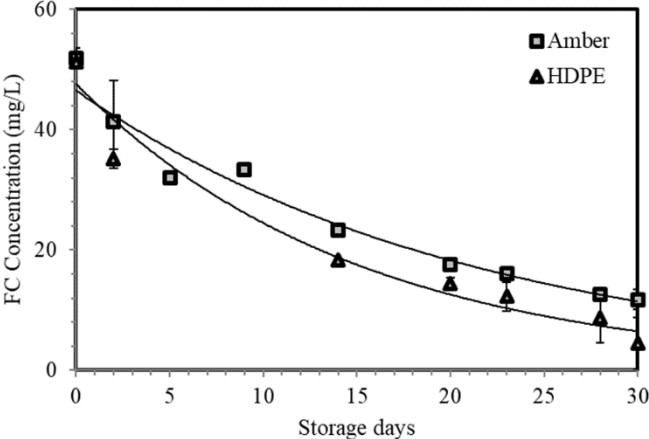

**Figure 6.** FC distribution of ESW stored in both bottles throughout the 30-day storage period.

The rate of FC decay of ESW during storage can be evaluated using Equation (10) [1,13].

$$\ln \frac{C_t}{C_o} = -kt \tag{10}$$

where:

$C_t$: FC concentration at any storage period $t$, mg/L
$C_o$: Initial FC concentration, mg/L
$k$: Rate of FC decay, day$^{-1}$
$t$: Storage period of ESW, day

The results are illustrated in Figure 7, and it is determined that stored ESW samples experienced single-stage first-order decay due to HOCl decomposition. The first-order decay was also observed in other studies [1,13,22]. Stored samples in amber glass bottles exhibited slower FC decay ($k = -0.046$ day$^{-1}$), compared to stored samples in HDPE bottles ($k = -0.066$ day$^{-1}$). Nonetheless, HDPE bottles could be used as a storage container due to the minimal difference between the two decay constants.

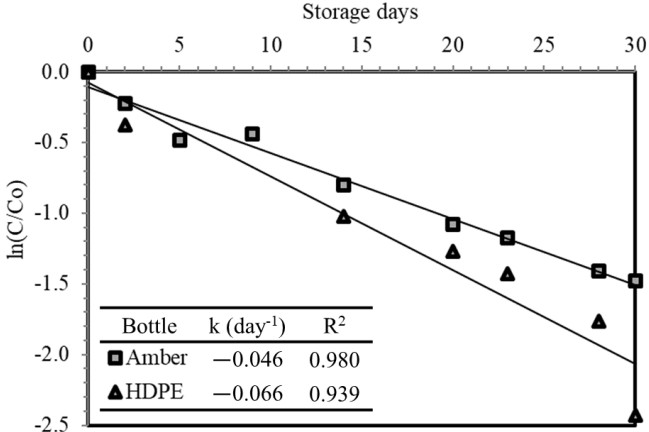

**Figure 7.** FC decay during the 30-day storage period.

### 4.2.2. Extension of 30-Day Storage Evaluation

The storage experiment was extended for two weeks to gather more information on the effect of storage conditions on the stability and disinfection performance. The remaining ESW in the 28-day-old bottles was used to evaluate the properties and bactericidal efficiency on days 42 and 49. From Table 1, it can be observed that there are minimal changes in the pH for both bottles. The oxidation-reduction potential of stored ESW in HDPE bottles had a minimal difference between days 28 and 42 but decreased suddenly to 632 mV by day 49. Meanwhile, FC in both bottles continued to decline until it reached less than 5 mg/L. The reason for the decrease in ORP and FC may be attributed to a decrease of hypochlorous acid and escape of chlorine gas through repeated opening of the bottles.

**Table 1.** pH, ORP, and FC concentration of ESW stored in both bottles at days 0, 28, 42, and 49.

| Container Type | Days | pH | ORP (mV) | Free Chlorine Concentration (mg/L) |
|---|---|---|---|---|
| Amber Glass | 0 | 4.21 | 1010 | 51.85 |
| | 28 | 6.70 | 853.50 | 12.67 |
| | 42 | 6.61 | 855 | 5.17 |
| | 49 | 6.71 | 815.50 | 2.95 |
| High-Density Polyethylene (HDPE) | 0 | 5.74 | 925 | 51.23 |
| | 28 | 7.24 | 797 | 8.77 |
| | 42 | 7.26 | 788.50 | 2.93 |
| | 49 | 7.30 | 623 | 0.99 |

### 4.3. Storage Effect on Disinfection Performance

The storage effect on the bactericidal efficiency of ESW on the fecal coliform *E. coli* was observed for 49 days. The freshly generated solution achieved complete inactivation of $1.5 \times 10^8$ CFU/mL of *E. coli* due to its slightly acidic (pH < 6.00) and highly oxidizing nature (ORP > 900 mV). It was hypothesized that the effectiveness of the disinfectant would decline through time due to changes in the physicochemical properties. However, no growth was detected for both bottles. A similar result was observed in another study where no surviving population was detected for 30 days, using acidic

and neutral electrolyzed water on *Salmonella typhii* and *Escherichia coli O157:H7* [2]. The main reason for this result could lie in the amount of hypochlorous acid present. Slow changes were observed throughout storage for ESW both in amber glass and HDPE bottles. From an initial solution containing 83.71% hypochlorous acid in amber glass bottles and 96.86% in HDPE bottles, the hypochlorous acid in 30-day-old solutions became 85.71%. The amount of hypochlorous acid is still high, and this could be enough to inactivate an average initial population of $1.5 \times 10^8$ CFU/mL. The 49-day-old ESW solutions have neutral pH and a lower oxidation-reduction potential (ORP = 815.5 mV for amber glass and ORP = 632 mV for HDPE bottles). Despite the low and almost negligible FC concentration, hypochlorous acid remains predominant for stored ESW in both bottles, since the pH of the solution is still below 7.5 [1,2,8].

## 5. Conclusions

The changes in the physicochemical properties affect the distribution of chlorine species in ESW stored in both amber glass and HDPE bottles. The increase in pH and decrease in ORP and FC concentration are due to a decreasing amount of hypochlorous acid through time. Equation (10) was used to investigate the FC decay of ESW through time. Therefore, it was observed that the decline was faster in HDPE bottles (k = −0.066 day$^{-1}$), compared to amber glass bottles (k = −0.046 day$^{-1}$). Further investigation revealed that instability in the physicochemical properties occurred in 49-day-old solutions stored in HDPE bottles. However, it did not affect the effectiveness of ESW as a disinfectant for the fecal coliform *E. coli*. No surviving population of *E. coli* was observed throughout the experimentation, and this could be due to the predominance of hypochlorous acid in the solution. Based on the results, the use of HDPE bottles for storage is recommended for 30 days only. Further investigation is needed for a full understanding on ESW's storage stability. The physicochemical properties and bactericidal efficiency may be evaluated by changing the storage conditions, such as storage time, temperature, and exposure to light and atmosphere. The antimicrobial assay can also be modified using different microbial species.

**Author Contributions:** All authors collaborated on this work. Conceptualization, A.B.B.; and R.G.D.; methodology, A.B.B.; R.G.D.; P.E.L.C.; and M.O.U.; formal analysis, A.H.O.; A.B.B.; M.O.U.; P.E.L.C.; and R.D; investigation, R.G.D.; writing—original draft preparation, R.G.D.; writing—review and editing, A.H.O.; A.B.B.; M.O.U.; P.E.L.C.; and R.G.D.; supervision, P.E.L.C.; and M.O.U.

**Funding:** This research was funded by the University Research Coordination Center of De La Salle University-Manila, grant number 56 N 3TAY 14–3TAY 15.

**Acknowledgments:** The authors would like to acknowledge the technical staff of the Microbiology Laboratory for their kind assistance and support.

**Conflicts of Interest:** The authors declare no conflict of interest.

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
