# Peer review of "Storage Stability and Disinfection Performance on Escherichia coli of Electrolyzed Seawater"

_water, doi:10.3390/w11050980_

Round 1

Reviewer 1 Report

The authors determined the effect of storage conditions (time, type of containers) of electrolyzed seawater on its stability (pH, ORP and free chlorine concentration) and bactericidal effectiveness on Escherichia coli. In preparatory studies they determined the optimal current and time of electrolysis. They indicated an increase in pH when reducing ORP and FC concentrations within 30 days. Nevertheless, hypochlorous acid remained the dominant component ensuring disinfectant action.

Comments:

Line 29-30 - Here, you can certainly give more references to the bibliography.

Line 31 - Here, in turn, as many as 7 references to the bibliography is unnecessary. I suggest you choose the most up-to-date and other quotes in other places of work.

Line 35 - Numbering of advantages is unnecessary. (Other 35 advantages of electrochemical disinfection include: on-site production, raw material availability, environmental-friendly process, less hazardous reaction, low cytotoxicity and no reported microbial resistance [4].)

Line 42 and 43 - I suggest to extend the references to the bibliography.

Line 56 – E. coli – in italic

Line 83 - How high was the ambient temperature? It was constant whether it was in the range. It should be completed.

Line 86 – E. coli - improve on: Escherichia coli

Line 89 – McFarland #5 [???]

Line 91 - no reference to the standard for the number of colonies of E.coli.

Line 100 - lack of ORP before 0,0 to 1000mV

Line 103 – why [1] was inserted?

In Fig. 2 should be: Cl2 and OCl-

Line 137 – should be…5.76, 3.77 and 3.35…

In Fig. 3, the diagrams b and c are the same, while the diagram c for free chlorine is missing. Thus, there is no representation of the discussion between lines 141 and 146.

In the signature of Fig. 3, A (1.5A) is missing.

In relation to which reference electrode was ORP measured? This information should be completed in the work (in chapter: Electrolysis Set up).

Why is standard deviation (?) marked only in some points in Figure 4a?

Line 170 and 181– Behavior of pH and ORP… (The pH and ORP distribution…/ Changes in pH and ORP of the electrolyzed…)

Line 177 and 178 - ppm convert to mg/L

Line 181 – (Fig 6.) The same signature as in Figure 5. Figure 6 applies to FC!

Line 184 and 185 - Improve on: C is the FC concentration at any storage period, Co is the initial free chlorine concentration, k is the rate of decay per day, and t is the storage period in days [1].

Line 187 – This behavior?

Authors often refer to [1]. There are many other studies in this area that would be worth quoting in this work. I suggest expanding the list of references. This one is very modest.

Line 205 – Table 1. Behavior?.....containers or bottles?

In table 1 - no unit for FC.

Line 218 – CFU – it should be should be explained in the text. The same applies to HDPE (when it was used in the text for the first time).

The conclusions should no longer refer to the bibliography.

In terms of academicity, the work is average, but it is valuable in practical terms. As the authors emphasize: The study can benefit people living near coastal areas where salt water is readily available. In addition, test results can be helpful in conditions such as sanitary disasters.

Author Response

Dear Reviewer,

Greetings!

We are submitting the revised manuscript entitled “Storage Stability and Disinfection Performance on Escherichia coli of Electrolyzed Seawater” (Water ID: 47974) to be considered for publication in MDPI Water. Your comments and suggestions are greatly appreciated. Revisions are thus incorporated to improve the content of the manuscript.  Please see attached file below for the  responses of the authors.

My co-authors and I look forward in receiving feedback from MDPI Water’s editorial board in due time. Thank you very much.

Regards,

Dr. Arnel B. Beltran

Associate Professor

De La Salle University, Manila

Reviewer 2 Report

Please add information about energy consumption based on 1 KL of treated water.

Methodology for bacteria deactivation is not clear, elaborate. 

What are the practical application of such technology?

Does this method produce hydrogen, split water? you need to consider? did you collect these gases? and why not?

The size of the mesh? hole size, any effect of this and the gap between the electrodes on the results?

Compare to UV/ efficiency and energy consumption.

Etc!

A lot missing from the article, please add more details to make things clear for the readers.

Author Response

(The authors gave the same response as above.)

Round 2

Reviewer 1 Report

After reading the revised manuscript and taking into account the responses to the comments, I support the publication of yours work.

However, please follow the line 146 (should be ... 5.76, 3.77 and 3.35). It is still ... 5.76, 3.77 and 3.35.

Reviewer 2 Report

Accept